# Fuzzy Cognitive Map Applications in Medicine over the Last Two Decades: A Review Study

**DOI:** 10.3390/bioengineering11020139

**Published:** 2024-01-30

**Authors:** Ioannis D. Apostolopoulos, Nikolaos I. Papandrianos, Nikolaos D. Papathanasiou, Elpiniki I. Papageorgiou

**Affiliations:** 1Department of Energy Systems, University of Thessaly, Gaiopolis Campus, 41500 Larisa, Greece; ece7216@upnet.gr (I.D.A.); npapandrianos@uth.gr (N.I.P.); 2Department of Nuclear Medicine, University Hospital of Patras, 26504 Rio, Greece; nikopapath@upatras.gr

**Keywords:** fuzzy cognitive maps, medicine, fuzzy logic, explainable artificial intelligence

## Abstract

Fuzzy Cognitive Maps (FCMs) have become an invaluable tool for healthcare providers because they can capture intricate associations among variables and generate precise predictions. FCMs have demonstrated their utility in diverse medical applications, from disease diagnosis to treatment planning and prognosis prediction. Their ability to model complex relationships between symptoms, biomarkers, risk factors, and treatments has enabled healthcare providers to make informed decisions, leading to better patient outcomes. This review article provides a thorough synopsis of using FCMs within the medical domain. A systematic examination of pertinent literature spanning the last two decades forms the basis of this overview, specifically delineating the diverse applications of FCMs in medical realms, including decision-making, diagnosis, prognosis, treatment optimisation, risk assessment, and pharmacovigilance. The limitations inherent in FCMs are also scrutinised, and avenues for potential future research and application are explored.

## 1. Introduction

Medical decision-making is a complex process that involves several factors, such as patient symptoms, medical history, test results, and clinical expertise [1]. The decisions that clinicians make significantly impact patient outcomes, making it a critical aspect of healthcare. In recent years, artificial intelligence (AI) has shown the potential to aid in medical decision-making, improving diagnosis accuracy and treatment planning [2,3]. 

AI has been making waves in the medical field in recent years. One of the significant advances has been in medical imaging [4]. AI algorithms can analyse medical images to identify anomalies that might be missed by human clinicians, improving the accuracy of diagnosis and treatment [4,5,6,7]. 

Fuzzy cognitive maps (FCMs) [8] are AI algorithms that can aid medical decision-making. FCMs are a mathematical tool that can model complex systems, such as the human body. FCMs can analyse patient data, such as symptoms, medical history, and test results, and visually represent the relationship between different variables [9]. FCMs can aid clinicians in making more informed decisions by providing a more comprehensive understanding of the patient’s condition [10]. The growing body of research on FCM applications in medicine has demonstrated their potential to improve healthcare decision-making and patient outcomes. As FCM modeling techniques evolve, their role in medical applications is expected to expand further, paving the way for more personalised and effective healthcare interventions.

This review paper intends to make a unique contribution to the study of FCMs in the medical domain by offering the first, to our knowledge, an extensive retrospective of literature published over the past two decades. By undertaking a systematic review of studies across various medical fields, including but not limited to decision-making, diagnosis, prognosis, risk assessment, treatment optimisation, and pharmacovigilance, this work fills a critical review gap that currently exists in this domain.

Our research has assembled, synthesised, and critically appraised diverse research strands exploring the use of FCMs, thus providing a consolidation of knowledge that has not been achieved previously. Moreover, our historical perspective allows for an examination of the evolution of FCM application in medicine, illuminating trends, patterns, successes, and challenges that have emerged over time. This temporal analysis enriches the understanding of the domain by situating current applications within a broader historical context.

Furthermore, we advance the domain by identifying and articulating the limitations of FCMs, a necessary effort for improving the robustness and reliability of its application in the future. By presenting these limitations, we offer valuable guidance for scholars and practitioners in managing the complexities embedded in FCM systems.

Lastly, we recognize the need to align the increasing body of literature with the latest methodological conclusions and directives. Therefore, this review concludes with an exploration of potential future directions for research and application. This foresight work sets a basis for contemporary investigations, helping researchers to navigate the evolving landscape of FCMs in the medical field and ultimately drive better health outcomes.

The remainder of the manuscript is organised as follows: Section 2 presents the methodology of this review, the research questions, the exclusion and inclusion criteria, and the review protocol. Section 3 presents the fundamental principles of FCMs, as well as their advantages and limitations. Section 4 presents the results of the literature review, highlighting significant works that underline applications of FCMs in the medical domain. Section 5 discusses the results and suggests future directions based on the determined limitations of the current body of literature.

## 2. Review Methodology

To execute the literature review, a procedural sequence encompassing planning, execution, and reporting phases has been implemented. During the planning phase, explicit research questions were formulated, and a review protocol was devised, delineating the targeted publication sources, search terminologies, and selection criteria. Subsequently, in the execution phase, the literature was amassed in adherence to an established review protocol. The selected literature underwent thorough analysis, involving extracting and synthesising pertinent data to address the defined research queries. Ultimately, this review outcomes were documented, aligning with the research questions and the systematic literature review’s overarching objectives.

### 2.1. Research Questions

The main objective of the present literature review is to present the applications of FCMs in Medicine over the last two decades. Hence, it furnishes insights into contemporary practices to leverage them for subsequent enhancements in the field. Consequently, the formulation of the ensuing research questions (RQs) has transpired:What are the leading medical entities covered by applications of FMCs?Over the analysis period, which medical problems or diseases have FCMs been associated with?What are the current state-of-the-art advancements and the limitations?

A concentrated methodology was adopted during the literature survey, with each article systematically examined to address the questions mentioned above. The accumulated data have been presented comprehensively to provide a holistic overview for a thorough understanding.

### 2.2. Review Protocol

#### Source, Terms, Inclusion, and Exclusion

Given the importance of ensuring the credibility and validity of sources in a medical review paper, we limited our search to PubMed, a database of biomedical literature curated by the National Center for Biotechnology Information (NCBI). PubMed’s rigorous selection process ensures that only peer-reviewed articles published in reputable journals are included, making it a trusted source of high-quality medical information. By focusing on PubMed articles, we sought to provide a comprehensive and reliable overview of current knowledge on the topic under investigation. PubMed’s extensive search capabilities enabled us to identify a wide range of relevant articles, ensuring we captured the spectrum of perspectives and findings on the subject. While other sources of medical information may exist, such as grey literature or non-peer-reviewed publications, we believe that PubMed’s rigorous selection process and extensive coverage make it the most appropriate resource for conducting a thorough and credible review paper. We used the keywords “Fuzzy Cognitive Maps” and “Fuzzy Logic” to find potential papers. Then, we reviewed the returned titles and abstracts to exclude works based on the following criteria:Articles published after the end of the year 2002.Original research articles (journals and conferences)Articles using any FCM, but not solely fuzzy logic.Articles demonstrating the use of the proposed methodology to solve a medical problem.

Our research review adheres to the PRISMA guidelines, ensuring a systematic and transparent approach to the selection, analysis, and reporting of relevant studies.

### 2.3. Literature Collection

The initial literature search returned 132 publications, excluding review articles. The topic-related publications were identified based on their title and abstracts. This process resulted in the exclusion of 87 publications. Figure 1 summarises the literature collection process.

A second screening was performed to identify articles that did not qualify based on the inclusion criteria. The latter operation resulted in the exclusion of one publication. Hence, the pool material for review included 42 papers. 

## 3. Fuzzy Cognitive Maps

### 3.1. Fundamentals of Fuzzy Cognitive Maps

FCMs are a mathematical modelling tool used for decision-making and knowledge representation. FCMs were introduced by Bart Kosko in 1986 [8] and have since been used in various fields such as engineering, economics, and social sciences [11].

The fundamental principle behind FCMs is the concept of “fuzziness” or uncertainty in the human thought process [12]. FCMs represent complex systems and their interrelationships by modeling the cognitive processes individuals use to make decisions [9]. This allows FCMs to account for ambiguity, imprecision, and subjectivity in decision-making, which is often impossible with traditional methods.

FCMs consist of a directed graph, with nodes representing concepts and edges representing causal relationships between them [9,13]. The nodes are associated with linguistic variables, which can take on fuzzy values. These fuzzy values indicate the degree of membership of a concept to a given set, typically represented by a membership function.

The weight of an edge in an FCM represents the strength of the causal relationship between the nodes it connects [9]. The weights are typically represented by linguistic variables, with fuzzy values indicating the degree of influence that one node has on another.

The activation of a node in an FCM represents the degree to which the concept it represents is present in the system [14]. The activation of a node is calculated by aggregating the weighted inputs from the nodes connected to using a transfer function. The transfer function is typically nonlinear, such as the sigmoid function, which maps the weighted inputs to a fuzzy value.

The activation of a node can be used to predict the system’s behaviour over time. By iteratively updating the activations of the nodes, it is possible to simulate the system’s dynamics and predict its future behaviour.

The complete process is described mathematically in the following section.

#### Mathematical Explanation

The activation of a node in an FCM is calculated as the weighted sum of the activations of the nodes that are connected to it using a transfer function [14]. Mathematically, the activation of node *i* is given by:(1)a(i)=T[∑(w(j,i)×a(j))]
where *w*(*j*,*i*) is the weight of the edge connecting node *j* to node *i*, and *T* is the transfer function.

The weights of the edges in an FCM are typically determined by experts in the relevant fields based on their knowledge and experience [15]. The weights can be represented using linguistic variables and fuzzy values, which indicate the degree of influence that one node has on another. For example, consider the following weight values:*w*(“temperature”, “crop yields”) = “strong positive”*w*(“sea level”, “coastal infrastructure”) = “moderate negative”*w*(“energy consumption”, “economic growth”) = “weak positive”

In this example, the weight between “temperature” and “crop yields” is a “strong positive” value, indicating that an increase in temperature would have a significant positive impact on crop yields. Similarly, the weight between “sea level” and “coastal infrastructure” is a “moderately negative” value, indicating that an increase in sea level would harm coastal infrastructure. The weight between “energy consumption” and “economic growth” is a “weak positive” value, indicating that there is a positive but relatively weak relationship between these two variables.

The transfer function in an FCM maps the weighted sum of the activations of the nodes to a fuzzy value, which represents the degree of activation of the node [16]. The most used transfer function is the sigmoid function, which is given by:(2)$$T(x)=1\;/\;(1+e^{-\alpha x})$$
where *α* is a scaling parameter that determines the steepness of the sigmoid function [16]. The output of the transfer function *T*(*x*) is a fuzzy value between 0 and 1, representing the node’s degree of activation.

The activations of the nodes in an FCM can be updated iteratively to simulate the system’s behaviour over time. The updated activation of node *i* at time *t* + 1 is given by:(3)a(i, t+1)=T[∑(w(j,i)×a(j, t))]
where *a*(*j*,*t*) is the activation of the node *j* at time *t*, and *w*(*j*,*i*) is the weight of the edge connecting node *j* to node *i*.

### 3.2. Recent Progress in FCMs

FCMs have emerged as a powerful tool for modelling complex systems, particularly those characterised by nonlinear dynamics, uncertainty, and multiple interacting elements [13]. Their ability to capture causal relationships, handle linguistic variables, and incorporate human expertise has made them valuable in various fields, including economics, finance, engineering, social sciences, and medicine. Recent research has focused on extending FCMs to address new challenges and broaden their applicability. One notable trend is the integration of intuitionistic theory, hesitancy theory, grey system theory, and wavelet theory. These extensions enhance FCMs’ ability to handle incomplete information, hesitancy, dynamic systems, and probabilistic fuzzy events, making them more versatile for modelling real-world scenarios.

Another area of active research is the development of improved learning algorithms for FCMs. Conventional FCMs require explicit expert input to specify the initial weights and connections between concepts. Recent approaches aim to automate this process, allowing FCMs to learn from data and self-organise their structure. This ability is important for applications with scarce or difficult-to-obtain expert knowledge.

Real-world systems often exhibit dynamic behaviour and are subject to external disturbances. Researchers are developing adaptive FCMs that can adjust their weights and connections in response to changing conditions. Additionally, methods for incorporating uncertainty into FCMs are being explored, allowing them to handle imprecise or noisy data more effectively.

### 3.3. From Theory to Real-World Scenarios

Creating an FCM involves a structured and iterative process encompassing problem definition, concept identification, causal relationship elicitation, initial state assignment, rule specification, FCM construction, learning algorithm selection, training and validation, model output and interpretation, and model refinement [17,18]. The pipeline (Figure 2) ensures that FCMs are designed effectively to represent the dynamics and relationships within complex systems, enabling accurate predictions, decision-making, and knowledge extraction. 

We can discretise the steps as follows:Clearly define the problem or system to be modelled using FCMs.Identify the fundamental concepts or variables that influence the system’s behaviour.Understand the causal relationships between these concepts, including positive and negative influences.Select a manageable number of concepts to avoid complexity and over-parameterisation.Gather knowledge and expertise from domain experts or data sources to understand the causal relationships between concepts.Represent these relationships as directed links in the FCM, using arrows to indicate the direction of influence.Assign weights to the links to represent the strength of the causal relationship.Assign initial states to each concept, representing their starting values or conditions.These initial states can be based on historical data, expert knowledge, or assumed values.Define linguistic rules that capture the qualitative relationships between concepts; these rules should express the conditional logic of the system’s behaviour.Use fuzzy logic to represent linguistic variables and their relationships.Build the FCM network by connecting nodes based on the identified causal relationships.Assign weights to the links, representing the strength of the causal influences.Implement the linguistic rules into the FCM’s structure.Choose an appropriate learning algorithm to update the FCM’s node states and weights.Adjust the learning parameters to ensure convergence and accurate model behaviour.Train the FCM using historical data or simulation scenarios.Monitor the model’s performance during training, evaluating its accuracy and stability.Simulate the FCM using the learned weights to predict future states or make decisions.Interpret the model’s output by analysing the changes in node states and the propagated causal influences.Consider the strength of causal links and the qualitative nature of linguistic rules for meaningful interpretation.Based on the training and validation results, refine the FCM’s structure, weights, and learning parameters.Adapt the model to incorporate new data, changing conditions, or emerging knowledge.

#### 3.3.1. Advantages and Limitations

FCMs combine the strengths of fuzzy logic and cognitive maps, offering a versatile framework for representing knowledge, simulating system behaviour, and making predictions [19].

One of the primary advantages of FCMs is their ability to handle linguistic variables. This makes them well-suited for modelling systems where precise quantitative data may be limited or difficult to obtain. In addition, FCMs can effectively represent the causal relationships between system components, capturing both positive and negative influences [17]. This enables them to capture the complex dynamics of real-world systems where interactions are often nonlinear and interdependent.

Another critical advantage of FCMs is their ability to be simulated. FCMs can be simulated using iterative algorithms to propagate changes through a network of nodes, enabling the prediction of future system states and the analysis of potential scenarios. This capability makes them valuable for decision-making and strategic planning. FCMs also provide a transparent and interpretable representation of system knowledge [20], allowing experts to understand the underlying relationships and reasoning behind model predictions. This facilitates the communication of complex system dynamics to stakeholders.

Finally, FCMs can be adapted to incorporate new information, changing conditions, or emerging knowledge, making them suitable for modelling dynamic and evolving systems [21]. This adaptability is particularly beneficial in situations where real-time data analysis is required.

Despite their many advantages, FCMs also have some limitations. One of the primary limitations is their dependence on expert knowledge [20]. FCMs rely heavily on expert knowledge to construct the network structure and assign weights to the connections. This can limit their applicability when specialist knowledge is scarce or impossible to obtain.

Another limitation is the sensitivity of FCM simulations to initial states. FCM simulations can be highly sensitive to the initial states of the nodes. This sensitivity can make achieving consistent and reliable predictions challenging, especially for complex systems with multiple interacting elements. Additionally, FCM simulations tend to diverge. FCM simulations can converge to stable states or diverge toward chaotic behaviour based on the initial states and the strength of causal influences. This divergence can hinder the interpretation of model results and limit their predictive accuracy.

Moreover, FCMs can have scalability issues. For large and complex systems, constructing FCMs with a manageable number of nodes and connections can be challenging. This can limit their applicability to real-world systems with numerous interacting elements. Finally, there is a lack of standardisation in FCM representations. No universally accepted standard for representing FCMs makes it difficult to compare and integrate results from different studies. This lack of standardisation can hinder the field’s development and adoption in broader applications.

Continued research aimed at addressing the limitations of FCMs, such as developing more robust learning algorithms, improving scalability, and establishing standards for representation, is expected to enhance their applicability further and expand their reach across various domains.

#### 3.3.2. The Role of FCMs in Explainable Artificial Intelligence

The European Union has passed new regulations requiring that people have a right to an explanation for algorithmic decisions that affect them [1]. This has increased the demand for explainability methods in machine learning and deep learning models.

One approach to making AI more explainable is to use FCMs [20,21]. In addition, FCMs are transparent, meaning that their internal structure can be easily visualised and analysed. This makes it easy to see how the model works and how it makes its decisions. FCMs are also informative, meaning that they can identify key factors that influence the system and simulate different scenarios and their potential outcomes. This can be helpful for decision-making. Finally, FCMs are transferable [20], meaning they can be adapted and applied to different systems by changing the nodes and edges in the model. This makes them a versatile tool for modelling a wide variety of systems.

## 4. Applications of FCMs in Medicine

The literature review identified seven categories of applications, namely, prognosis and prevention, risk assessment, treatment planning, policymaking, healthcare services, and ethics (Figure 3).

Moreover, based on the problem being solved by these systems, the following diseases or conditions are associated: COVID-19, coronary artery disease, dengue, depression, obstetrics, chronic obstructive pulmonary disease, stroke, plastic surgery, aesthetics, breast cancer, diabetes, radiation therapy, pulmonary infections, uncomplicated urinary tract infection, language impairment, dysarthria, apraxia of speech, gastric cancer, chronic nerve diseases, esophageal cancer, body dysmorphic disorder, protein function prediction, and chronic obstructive pulmonary disease (Figure 4). A summary of the publications can be found in Table 1.

### 4.1. Medical Diagnosis

FCMs have been broadly used for diagnostic purposes in multiple medical domains. By mapping the relationships between variables, such as symptoms, risk factors, and biomarkers, FCMs can help healthcare providers make accurate diagnoses. FCMs can be trained on large datasets to identify patterns and associations that may not be immediately apparent to a human observer. This can help improve the accuracy and efficiency of disease diagnosis, leading to better patient outcomes.

Amirkhani et al. [22] addressed the challenge of annotating protein-DNA interactions. They proposed a novel approach using the FCM model to predict DNA-binding residues within local segments of protein sequences. The FCM model utilised information such as putative solvent accessibility, evolutionary conservation, and the relative propensities of amino acids to interact with DNA to identify potential DNA-binding residues. Empirical testing on a benchmark dataset shows that the FCM model achieved an AUC (Area Under the Curve) of 0.72, surpassing the performance of the hybridNAP predictor and various popular machine learning methods like Support Vector Machines, Naive Bayes, and k-Nearest Neighbor. The paper also demonstrates that employing a short sliding window further enhances the model’s predictive quality.

Al-Halabi et al. [23] delved into the cognitive aspects of surgical competence, particularly in plastic surgery procedures, namely breast augmentation and flexor tendon repair. Recognising cognition as a vital component of surgical proficiency, the research aimed to analyse and compare the mental models employed in these two distinct procedures. The approach involved generating task lists based on cognitive task analyses, literature reviews, and field observations. Two FCMs were then developed to visually represent and analyse the cognitive processes inherent in each procedure. By comparing these models and drawing insights from the literature, the study identified five cognitive competency domains relevant to plastic surgery: situation awareness, decision-making, task management, leadership, and communication and teamwork. The research also highlighted differences in decision-making processes between elective and trauma settings.

Apostolopoulos et al. [10] proposed a Computer-Aided Diagnostic model for precisely diagnosing Coronary Artery Disease (CAD). The methodology is based on State Space Advanced FCMs (AFCMs), an evolution of traditional FCMs. The model incorporated a rule-based mechanism for increased knowledge and interpretability. Evaluations using a CAD dataset from the University Hospital of Patras, Greece, showed the effectiveness of the AFCM approach, achieving 85.47% accuracy in CAD diagnosis, a 7% improvement over the traditional FCM approach.

Feleki et al. [24] introduced a novel and transparent model called DeepFCM for diagnosing CAD using Myocardial Perfusion Imaging (MPI) and clinical data. DeepFCM combines an image classification—Convolutional Neural Network (CNN)—with an FCM-based classifier for integrating clinical data. The model incorporates expert knowledge for initialising interconnections and employs Particle Swarm Optimization (PSO) to adjust weights. Critical features for explainability include Gradient Class Activation Mapping (Grad-CAM) for highlighting significant regions on images, disclosure of internal weights and their impact, and utilisation of Generative Pre-trained Transformer (GPT) for generating meaningful explanations. The model achieved an accuracy of 83.07%, a sensitivity of 86.21%, and specificity of 79.99%. The proposed framework enhances clinical interpretability and can be applied in daily routines and educational settings.

Hoyos et al. [25] proposed a system for diagnosing dengue (break-bone fever) based on its severity. The system utilised an FCM, incorporating clinical and laboratory variables associated with dengue. The model demonstrates a classification accuracy of 89.4%, indicating its effectiveness in diagnosing dengue and evaluating the behaviour of related variables.

Another study by Sovatzidi et al. [26] addressed the prevalent and impactful issue of depression, particularly among adolescents, by introducing a novel framework for automatically generating FCMs. The proposed Constructive FCM (CFCM) leverages electroencephalogram (EEG) data to assess the severity of depression. By applying a Constructive Fuzzy Representation Model (CFRM), CFCM identifies causal relationships between brain activity and depression more intuitively. By reducing reliance on expert input and minimising manual interventions, CFCM offers a streamlined approach to FCM construction. Additionally, it incorporates a built-in mechanism for dimensionality reduction, ensuring interpretability in decision-making while remaining aware of uncertainties and maintaining simplicity in implementation. Experimental results conducted on a recent publicly available dataset validate the efficacy of the proposed framework and underscore its advantages.

Georgopoulos et al. [27] introduced a novel hybrid modelling methodology for medical diagnosis decisions, extending competitive FCMs with genetic algorithms for enhanced concept interaction. The synergy of these methodologies was achieved through a newly proposed algorithm, resulting in more reliable advanced medical diagnosis support systems. The technique was successfully applied to model and test a differential diagnosis problem in speech pathology for diagnosing language impairments. This approach proved effective, particularly when decisions were not distinct.

Papageorgiou et al. [28] developed an advanced diagnostic method for grading urinary bladder tumours using a novel soft computing modelling methodology. The approach combined FCMs with the unsupervised active Hebbian learning (AHL) algorithm. Histopathological features for tumour grading were defined by expert histopathologists, resulting in a nine-concept FCM model. The AHL algorithm was applied to enhance the FCM model’s classification ability. The proposed method achieved a classification accuracy of 72.5%, 74.42%, and 95.55% for tumours of grades I, II, and III, respectively. The technique offers a transparent and explicative approach to support tumour-grade diagnosis decisions.

The same research team enhanced the proposed method in [29]. The evaluation was also conducted using more data (571 participants) via 10-fold cross-validation, achieving 77.95% accuracy, 76.98% sensitivity, 77.39% specificity, and 73.97% precision. The results demonstrate the efficiency of the proposed model, outperforming traditional machine learning algorithms.

### 4.2. Medical Decision Support Systems

FCMs play an essential role in advancing Medical Decision Support Systems (MDSS) and enhancing the capabilities of healthcare professionals to make well-informed and timely decisions. By modelling the intricate relationships between clinical parameters, patient history, and treatment options, FCMs offer a dynamic framework for MDSS. Healthcare practitioners can leverage FCMs to analyse complex medical scenarios, considering multiple variables and their interdependencies. This not only aids in diagnosing challenging cases but also facilitates personalised treatment recommendations based on individual patient profiles. The inherent adaptability of FCMs to handle uncertainty and imprecision aligns seamlessly with the nuanced nature of medical decision-making, making them invaluable tools in developing and improving MDSS. FCMs contribute to the evolution of healthcare practices, fostering a more intelligent and responsive approach to medical decision support.

Stylios and Georgopoulos [15] proposed the basic principles and functionalities of an MDSS applied to obstetrics to determine whether they should proceed with a Caesarian section or a natural delivery based on the physical measurements. Although simulations or real-world data were absent from the study, the proposed MDSS is feasible since its concepts correspond to the everyday routine.

Georgopoulos et al. [30] introduced a new methodology to extend the application of FCM-based MDSSs for learning and educational purposes, employing a scenario-based learning (SBL) approach. This is particularly significant in medical education as it allows aspiring medical professionals to safely explore various “what-if” scenarios through case studies, preparing them for handling critical adverse events effectively.

Another study [31] by Papageorgiou et al. focused on medical knowledge representation and reasoning using probabilistic and fuzzy influence processes in the semantic web to support decision-making tasks. Bayesian belief networks (BBNs) and FCMs are dynamic influence graphs for formalising medical knowledge. The study introduced an MDSS that uses a general-purpose reasoning engine, EYE, with the necessary plug-ins to perform reasoning on these knowledge models. The proof-of-concept example, urinary tract infection (UTI), was chosen to examine the proposed formalisation techniques. Medical guidelines for UTI treatment were formalised into BBN and FCM knowledge models. An evaluation of 55 patient cases demonstrated that the suggested approaches efficiently formalise medical knowledge in the semantic web, providing front-end decisions on antibiotic suggestions for UTI.

Lucchiari et al. [32] explored diagnostic reasoning in clinical performance, emphasising the impact on care quality and safety. While traditional models assume objective and logically driven information in diagnostic reasoning, the actual diagnostic process is influenced by cognitive biases. Although debiasing techniques aim to address these biases, their implementation in clinical practice faces challenges. The authors presented a conceptual scheme for the diagnostic process using FCMs, highlighting the need for balanced models incorporating cognitive and technological factors to improve diagnostic accuracy and safety in healthcare.

### 4.3. Prognosis and Prevention

The predictive capability of FCMs empowers early intervention and preventive measures, ultimately improving patient outcomes. The adaptability of FCMs to handle uncertainty and imprecision in medical data makes them particularly valuable for understanding the intricate dynamics of diseases, aiding healthcare providers in making informed decisions to enhance prognosis and prevent the onset or progression of illnesses.

Wu et al. [33] focused on addressing the complexity of factors affecting the health of older adults in rural China. The authors introduced an extended probabilistic linguistic FCM model. The model effectively deals with uncertainty, reflecting diverse expert opinions. The research identified education as the most critical factor influencing rural older adults’ health, followed by previous occupational experiences, psychology, and physical exercise. The study also emphasised the importance of considering various factors and recommended tailored health interventions for improved and sustainable rural elderly health in China.

Khodadadi et al. [34] proposed an application of FCMs for diagnosing the risk of ischemic stroke. The nonlinear Hebbian learning method was employed for training FCMs. The proposed method determines individual risk rates based on neurologists’ opinions. Through 10-fold cross-validation with 110 real cases, the model’s accuracy was assessed and compared using a support vector machine and K-nearest neighbours. The proposed system exhibits superior performance, achieving a total accuracy of 93.6 ± 4.5%.

Finally, Najafi et al. [35] performed a cross-sectional study in Iran, specifically in Zanjan Province, to explore the hypothesis that food insecurity may contribute to esophageal cancer among women. The research involved 580 women aged 40–70, with 150 having esophageal cancer. FCMs were employed for the analysis. The study revealed food insecurity rates, including hunger and hidden hunger, and assessed the accuracy of a questionnaire for screening hunger. The findings suggested an association between food insecurity, body mass index (BMI), and esophageal cancer, with increased rates of underweight and decreased rates of overweight and obesity in the study population.

### 4.4. Risk Assessment

In medical risk assessment, FCMs serve as powerful tools for comprehensively analysing and evaluating the multifaceted aspects contributing to health risks. By integrating diverse data sources and considering the interplay of variables, FCMs provide a holistic approach to risk assessment. This enables healthcare professionals to identify potential threats to patient well-being, assess the likelihood of adverse events, and implement targeted interventions to mitigate risks.

Subramanian et al. [36] addressed the increasing demand for classifying women into different risk groups for developing breast cancer (BC). The focus was on developing an integrated risk prediction model using a two-level fuzzy cognitive map (FCM) approach. The level-1 FCM modelled the demographic risk profile, employing a nonlinear Hebbian learning algorithm to predict BC risk grades based on demographic factors. These predictions were validated using established BC risk assessment models (Gail and Tyrer–Cuzick). The level-2 FCM modelled screening mammogram features, using a data-driven Hebbian learning algorithm to predict BC risk based on mammographic image characteristics. The integrated model overcomes the limitations of existing models by considering both demographic factors and mammogram findings, providing risk predictions in qualitative grades. The proposed model’s predictions align with the Tyrer–Cuzick model for many cases, while its overall accuracy in tumour grading is 94.3%. The proposed model demonstrates superior testing accuracy using a 10-fold cross-validation technique compared to other standard machine learning models.

Mahmoodi et al. [37] developed a framework for assessing the risk of gastric cancer (GC) using FCMs. Known for their strength in complex system modelling, FCMs were employed in the system, utilising the Nonlinear Hebbian Learning (NHL) algorithm. Medical records from 560 patients at Imam Reza Hospital in Tabriz City were used, with 27 compelling features in gastric cancer selected based on expert opinions. The proposed method achieved a prediction accuracy of 95.83%, outperforming other decision-making algorithms such as decision trees, Naïve Bayes, and an Artificial Neural Network (ANN). The system was deemed simple, comprehensive, and more effective for assessing GC risk, providing healthcare professionals with a valuable tool for predicting risk factors in the clinical setting.

Papageorgiou et al. [38] addressed the need for individualised risk assessment for familial breast cancer (FBC) among women aged 40–49 with a family history. The proposed FCM employed NHL to learn causal weights from 40 patient records, achieving a diagnostic accuracy of 95%. The results align with the Tyrer–Cuzick model for 38 out of 40 patient cases (95%), outperforming standard risk evaluation tools like Gail and NSAPB models. The proposed model also demonstrates higher accuracy in identifying high-risk women than traditional models and outperforms other machine learning-based inference engines and previous FCM-based risk prediction methods for breast cancer.

### 4.5. Treatment Planning

FCMs are instrumental in revolutionising treatment planning in the medical domain. FCMs offer a dynamic framework for optimising therapeutic approaches. FCMs’ capacity to handle ambiguity and incomplete information makes them particularly valuable in the complex and evolving landscape of treatment planning, fostering a more adaptive and patient-centric healthcare paradigm.

Giles et al. [39] employed FCMs to represent and compare Canadian aboriginal and conventional science perspectives on the determinants of diabetes. The analysis drew from published articles in Medline and PubMed (1966–2005). FCMs enabled a detailed description of the complex system involving culture, spirituality, and balance that underlies the aboriginal view of diabetes. It also highlighted the potential to identify more concrete stressors and outcomes from these less tangible factors, making them manageable and monitorable. This preliminary comparison illustrated FCM’s capability to extract, compare, and integrate perspectives from different knowledge systems, providing health management and policy development insights.

In another study by Papageorgiou [40], the complex task of determining appropriate antibiotics and treatment for uncomplicated urinary tract infection (uUTI) using a medical decision-making model was addressed. FCMs were proposed as an innovative and flexible approach to handle uncertainty and missing information in this context. The FCM-uUTI DSS software tool (default version) was introduced as a decision support system for uUTI treatment management. The tool, tested on 38 patient cases, demonstrated functionality and reliability, providing front-end decisions on antibiotic suggestions for uUTI treatment. The results highlighted the tool’s potential as a helpful reference for physicians and patients, with straightforward graphical representation and simulation capabilities, making medical knowledge widely accessible through computer consultation systems.

### 4.6. Policymaking

In the context of healthcare policymaking, FCMs provide a sophisticated tool for understanding the interconnected elements influencing public health decisions. FCMs facilitate the formulation of evidence-based policies that address the diverse needs of populations. Policymakers can utilise FCMs to model the potential impacts of different interventions, ensuring more informed and robust decision-making processes. The inherent flexibility of FCMs allows for integrating diverse perspectives and uncertainties, offering a comprehensive approach to policy development that considers the nuanced and evolving nature of healthcare systems.

For example, Babroudi [41] et al. addressed the challenges faced by hospitals during the COVID-19 pandemic, emphasising the importance of maintaining high-quality healthcare services. Evaluating the service performance SERVPERF standard criteria, typically designed for normal circumstances, the study adapts them to the context of infectious disease spread, such as the COVID-19 pandemic. Using Z-Number theory and FCMs, the causal relationships between criteria were analysed to determine their importance in the prevalence of five infectious diseases. The results indicated that hospital reliability, hospital hygiene, and completeness of the hospital are the most influential criteria in improving the quality of health services during infectious disease outbreaks. This research contributes to the literature by comprehensively analysing and prioritising criteria to measure health service quality in the context of infectious disease spread.

In another example, Dogu et al. [42] presented an innovative approach that integrates statistical-based FCMs (SBFCM) with ANNs to predict the length of hospital stay for chronic obstructive pulmonary disease (COPD) patients experiencing acute exacerbation. The SBFCM method is designed to identify input variables for the ANN model, employing statistical analysis to gather initial information for experts and subsequently incorporating expert opinions to construct a conceptual map of the system. Combining SBFCM and ANN methods enriches the prediction model with statistical data and expert insights. In numerical applications, the proposed approach demonstrated superior performance to conventional methods and other machine learning algorithms, achieving an accuracy of 79.95%. This outcome emphasises the efficacy of involving expert opinions in medical decision-making. The study contributes to establishing a medical decision support framework, enhancing the prediction of hospital stay duration, and facilitating more effective hospital management.

### 4.7. Services and Ethics

Fuzzy Cognitive Maps (FCMs) significantly optimise healthcare services by modelling the intricate relationships within healthcare systems. These maps represent the interactions among various components, such as patient flow, resource allocation, and service delivery. By leveraging FCMs, healthcare providers can enhance operational efficiency, streamline processes, and improve the overall quality of care. FCMs’ adaptability to changing conditions and their capacity to account for uncertainties make them invaluable tools for designing resilient and responsive healthcare services. From hospital management to patient care coordination, FCMs offer a versatile approach to enhancing the delivery of healthcare services in a rapidly evolving medical landscape.

An example comes from a paper focused on a hospital problem in Brazil. Poleto et al. [43] mention that Brazil’s hospital organisations have embraced telehealth systems to extend healthcare services to populations facing limited access, primarily due to geographical distances between communities and hospitals. The significance and utilisation of these services have recently surged mainly due to COVID-19-related mobility interventions at the state level. These telehealth systems handle sensitive and confidential data, including medical records, medication prescriptions, and diagnostic results. Recognising the impact of cybersecurity on the development of telehealth strategies is vital for establishing secure systems for day-to-day operations. In the context presented in this article, FCMs were employed to distill the intricacies of cybersecurity in telehealth services into comprehensible and objective results within an expert-based cognitive map. This tool facilitated the construction of scenarios that simulated potential implications arising from common factors affecting telehealth systems. FCMs contribute to a heightened understanding of cybersecurity strategies through expert knowledge and scenario analysis, fostering the evolution of cybersecurity practices in telehealth services.

In medical ethics, FCMs offer a unique perspective by providing a structured framework for navigating the complexities of moral and ethical considerations in healthcare.

Meier et al. [44] proposed leveraging machine intelligence to address real-life moral dilemmas in clinical settings. They introduced METHAD, a framework that systematically breaks down medical ethics cases into quantifiable parameters, offering a computerised approach to modelling their assessment. The study involved selecting an underlying moral theory, exploring technical solutions, and advocating for FCMs as an ideal tool for constructing an ethical advisory system. The authors presented the algorithm’s performance and acknowledged its limitations. Despite the technological capability, the paper questioned the ethical implications of implementing such a system in clinical practice. While autonomous vehicles making moral decisions are more acceptable to people than machine intelligence in clinical settings, the authors emphasised the importance of human contact in medicine. They argued that the intimate relationship between patients and medical personnel, guided by empathy rather than computerised calculations, is integral to ethical decision-making.

The authors foresee ethical advisory systems supporting, rather than replacing, human judgment. The paper encouraged starting a societal discussion on using machine intelligence in ethical decision-making. It stressed the need to consider the advantages and disadvantages of these emerging options, recognising the resistance to entrusting patients’ fate to non-biological systems and highlighting the importance of ethical deliberation as technology advances.

**Table 1 bioengineering-11-00139-t001:** Summary of reviewed publications.

Publication	Year	Domain	Associated Medical Entity
Feleki et al. [24]	2023	Diagnosis	Coronary Artery Disease
Feleki et al. [29]	2023	Diagnosis	Coronary Artery Disease
Al-Halabi et al. [23]	2022	Diagnosis	Plastic Surgery and Aesthetics
Hoyos et al. [25]	2022	Diagnosis	Dengue
Sovatzidi et al. [26]	2022	Diagnosis	Depression
Sarmiento et al. [45]	2021	Diagnosis	-
Apostolopoulos et al. [10]	2021	Diagnosis	Coronary Artery Disease
Apostolopoulos et al. [46]	2020	Diagnosis	Coronary Artery Disease
Amirkhani et al. [22]	2018	Diagnosis	Protein-DNA
Lucchiari et al. [47]	2014	Diagnosis	Seizures
Douali et al. [48]	2014	Diagnosis	Urinary Tract Infection
Lee et al. [49]	2012	Diagnosis	Pulmonary Infections
Georgopoulos et al. [27]	2009	Diagnosis	Language Impairment
Nguyen et al. [50]	2008	Diagnosis	Protein Functions
Papageorgiou et al. [28]	2006	Diagnosis	Tumour grading
Georgopoulos et al. [51]	2005	Diagnosis	Dysarthria and Apraxia of Speech
Georgopoulos et al. [52]	2003	Diagnosis	Language Impairment
Meier et al. [44]	2022	Ethics	-
Poleto et al. [43]	2021	Healthcare Services	-
Papageorgiou et al. [53]	2011	Healthcare Services	Pulmonary Infections
Georgopoulos et al. [30]	2014	MDSS	Labor
Papageorgiou et al. [31]	2013	MDSS	Urinary Tract Infection
Lucchiari et al. [32]	2011	MDSS	-
Stylios et al. [15]	2010	MDSS	Obstetrics
Papageorgiou et al. [54]	2007	MDSS	-
Papageorgiou et al. [55]	2006	MDSS	Bladder Tumor
John et al. [56]	2005	MDSS	Flu
Babroudi et al. [41]	2021	Policymaking	COVID-19
Groumpos et al. [57]	2021	Policymaking	COVID-19
Dogu et al. [42]	2021	Policymaking	Chronic Obstructive Pulmonary Disease
Saul et al. [58]	2022	Prevention	Healthy Habits
Wu et al. [33]	2022	Prevention	-
Khodadadi et al. [34]	2019	Prevention	Stroke
Najafi et al. [35]	2018	Prevention	Esophageal cancer
Billis et al. [59]	2014	Prevention	Geriatric depression
Mahmoodi et al. [37]	2020	Risk assessment	Gastric Cancer
Subramanian et al. [36]	2015	Risk assessment	Breast Cancer
Papageorgiou et al. [38]	2015	Risk assessment	Breast Cancer
de Brito et al. [60]	2013	Risk assessment	Body Dysmorphic Disorder
Papageorgiou et al. [40]	2012	Treatment Planning	Urinary Tract Infection
Giles et al. [39]	2007	Treatment Planning	Diabetes
Papageorgiou et al. [61]	2003	Treatment Planning	Radiation Therapy

## 5. Discussion and Conclusions

### 5.1. Summary of Findings

Applying FCMs in medical diagnosis involves modelling intricate relationships among symptoms, patient history, and diagnostic indicators. FCMs offer a dynamic framework for healthcare professionals to navigate the complexity of diagnostic decision-making, facilitating a more sophisticated and adaptive approach. By capturing uncertainties inherent in medical data, FCMs enhance diagnostic accuracy and enable personalised patient care. Table 2 summarises the key findings and contributions of the presented literature.

One of the most recurrent themes is the FCM’s application in diagnosing CAD. The DeepFCM [24] model integrated MPI, clinical data, and natural language insights, achieving an accuracy rate of 83.07%. However, a similar application [29] that introduced the FCM-PSO and DeepFCM for CAD classification, yielded a slightly lower accuracy of 77.95%. However, this study presented enhanced explainability. The minor inconsistency in accuracy across these two studies suggests that the model’s structure and inputs can influence performance, reflecting the FCM’s sensitivity to these parameters.

The deployment of FCM in assessing cognitive competencies in medical processes is another intriguing trend. FCM’s application in evaluating cognitive competency domains for breast augmentation and flexor tendon repair in plastic surgery, inherently varied from disease diagnosis, thus illustrating the model’s versatility. These cognitive assessments can potentially transform medical education paradigms and foster the development of more comprehensive practitioner evaluation systems.

In infectious disease diagnosis, the use of FCM has shown impressive results. A clinical decision-support system for dengue diagnosis [25], built on signs, symptoms, and laboratory tests, yielded an accuracy of 89.4%. The high accuracy obtained underscores the potential of the FCM in this domain and hints at its applicability in managing future infectious disease outbreaks.

Another cultural discrepancy comes from the application of FCM in Northern Nigeria, which identified the major causes of short birth intervals [45]. This highlights the FCM’s utility in sociocultural investigations and its potential in developing culturally sensitive healthcare interventions. Simultaneously, it points towards the model’s need to account for varying cultural factors effectively.

Comparatively, some studies fall short, like the FCM model proposed for predicting DNA-binding residues in protein sequences that only achieved an Area Under the Curve (AUC) of 0.72 [22]. This inconsistency underlines the fact that the setup and parameterization of the FCM are critical to its effectiveness, which, if neglected, could limit its applicability in certain domains.

The use of FCMs extends beyond traditional applications, addressing diverse domains like cognitive competency analysis, sociocultural health investigations, and ambiguous neurological contexts [41,62,63,64,65,66,67,68,69,70,71,72,73]. The versatility of FCM perception can cater to the exhaustive demands of such varied fields, underlining its scenario adaptability and nuanced perception of perceived complexities.

On a revolutionary note, FCMs have progressed into ethical decision-making and cybersecurity strategies in telehealth services. This move from a purely medical domain toward more interdisciplinary fields showcases the model’s adaptive nature, apt for the evolving needs of the medical field.

Additionally, FCMs have made significant contributions to predictive modeling, with applications in patient state forecasting and prediction of pandemic trends. Their dynamic modelling ability triumphs over traditional statistical models, laying the groundwork for early interventions and preemptive actions.

Finally, the role of FCMs in risk prediction and decision support for aiding the aging population further broadens their usability. Their performance in detecting ailments such as depression emphasises their potential in supporting healthcare management for aging well.

### 5.2. FCM Contributions in the Medical Domain

FCMs play a crucial role in prognosis and prevention by modelling complex relationships among health factors [17,74,75]. They help healthcare professionals predict disease progression, identify risk factors, and implement preventive measures. The adaptability of FCMs to handle uncertainty contributes to improved patient outcomes through early intervention and personalised preventive strategies.

In risk assessments, FCMs provide a holistic approach by integrating diverse data sources and considering the interplay of variables. This allows healthcare professionals to identify and mitigate potential threats to patient well-being. FCMs’ capability to capture uncertainties enhances the accuracy of risk assessment, contributing to more personalised and effective risk management strategies in healthcare.

Indeed, FCMs revolutionise treatment planning by modelling relationships between symptoms, patient characteristics, and treatment modalities. Healthcare practitioners leverage FCMs to tailor treatment plans based on individual patient profiles, optimising therapeutic approaches and improving overall efficacy. The adaptability of FCMs to handle ambiguity aligns with the evolving nature of treatment planning in healthcare.

In addition, FCMs contribute to healthcare policymaking by providing a sophisticated tool to understand interconnected elements influencing decisions. Policymakers can model the potential impacts of interventions, ensuring evidence-based and robust decision-making. FCMs’ flexibility allows the integration of diverse perspectives, fostering a comprehensive approach to policy development considering healthcare systems’ nuanced and evolving nature.

FCMs optimise healthcare services by modelling relationships within healthcare systems, offering a dynamic representation of interactions among components such as patient flow and resource allocation. FCMs enhance operational efficiency and improve care quality, providing a versatile approach to healthcare service enhancement in a rapidly evolving medical landscape.

While the number of published papers has remained stable in recent years (Figure 5), the urgent need for explainable artificial intelligence and the increasing need for prevention medicine and risk assessment may catalyse the scientific efforts for adopting FCM methods for problem-solving.

### 5.3. Performance Metrics

Our review of the available literature has indicated a range of metrics employed to measure the effectiveness of FCMs for various applications in medicine, and this breadth of metrics may pose a challenge for meaningful comparative evaluation. Predominantly, studies utilised measurements like accuracy, sensitivity, specificity, the area under the AUC score, and the F1 score [60,76,77,78,79,80,81,82,83,84,85,86,87,88,89,90,91,92,93,94,95,96,97,98,99,100,101,102,103,104,105]. These metrics provide valuable insights into model performance; however, their inconsistent use can precipitate difficulties in standardised comparison between different studies.

Observations also confirmed the rarity of certain essential metrics in many studies. For instance, although confusion matrix-based metrics are crucial for evaluating diagnostic models, especially in instances of class imbalance, they are conspicuously absent in a significant number of works. Similarly, other class imbalance-conscientious metrics, such as Matthews Correlation Coefficient and Balanced Accuracy, are not frequently reported.

Another concerning observation is the ambiguity surrounding the benchmark against which the FCM model’s performance is evaluated. It is, at times, unclear if the model’s agreement is with the medical experts who identified the true labels in the classification problem or with an established golden standard, thus causing potential confusion and possible misinterpretation of results.

Furthermore, many simulation studies, although crucial for demonstrating the feasibility of FCMs, often do not report quantifiable metrics. The lack of these metrics can impede systematic assessment and understanding of the FCM’s performance trends in simulated environments.

Additionally, problems that involve revealing the synergies between concepts generally do not require metrics and are, thus, not directly comparable to other studies. This particular conundrum indicates the intrinsic versatility of FCMs and the resulting complexity in evaluating their effectiveness across varied applications.

However, these concerns about inconsistent metric use do not exclusively reside within FCMs; they reflect a broader challenge seen across the medical domain. There is increasing recognition of the need for more standardised, transparent, and replicable metrics to allow for better comparison, replication, and meta-analyses across all medical studies. Moving toward a unified set of evaluation metrics and benchmarks for FCMs would not only support the medical field in harnessing the potential of FCMs more effectively but also contribute to the broader movement seeking to improve replicability and transparency in medical research.

### 5.4. Future Directions

In advancing the future utility of FCMs, researchers must dedicate attention to elucidating the explanations provided by FCMs. While FCMs offer a valuable framework for modelling complex relationships, the interpretability of their outcomes remains crucial for practical implementation. The weight matrix within FCMs holds substantial discussion capacity, warranting in-depth investigation into its role and implications. Researchers should explore how individual weights contribute to the overall decision-making process, enhancing transparency and facilitating a more profound understanding of FCM-generated insights.

A critical avenue for future research involves actively addressing the inherent limitations of FCMs, particularly non-dynamism, without resorting to mirroring neural network architectures [106,107,108,109,110,111]. While FCMs provide a structured approach to modelling uncertainties, their static nature poses challenges in capturing dynamic real-world systems. Researchers are urged to explore innovative methodologies that preserve FCMs’ transparency while introducing dynamic elements. This pursuit aims to fortify the adaptability of FCMs without sacrificing their inherent interpretability, thus overcoming a pivotal challenge in their applicability.

An essential aspect of the evolution of FCMs lies in the rigorous evaluation of model performance by scrutinising external datasets. Presently, many FCM studies lack this crucial step, limiting the generalizability of their findings. Future research should prioritise the assessment of FCMs on diverse datasets beyond those used in their development, thereby elucidating the transferability of these models across varying contexts. Such an approach ensures robustness and reliability, offering a more comprehensive understanding of the efficacy and limitations of FCMs in real-world applications.

To augment the versatility of FCMs in diverse domains, there is a pressing need to extend their functionalities to accommodate various modalities, including but not limited to medical images and multi-modal inputs. Current FCM applications predominantly focus on symbolic data, and incorporating diverse modalities would enhance their applicability in fields such as medical imaging analysis. Researchers should actively explore methodologies to seamlessly integrate and process information from disparate sources within the FCM framework. This expansion aims to broaden the scope of FCMs, enabling them to model and analyse complex systems characterised by diverse data types effectively, thereby contributing to their efficacy in addressing multifaceted challenges across numerous domains.

## Figures and Tables

**Figure 1 bioengineering-11-00139-f001:**
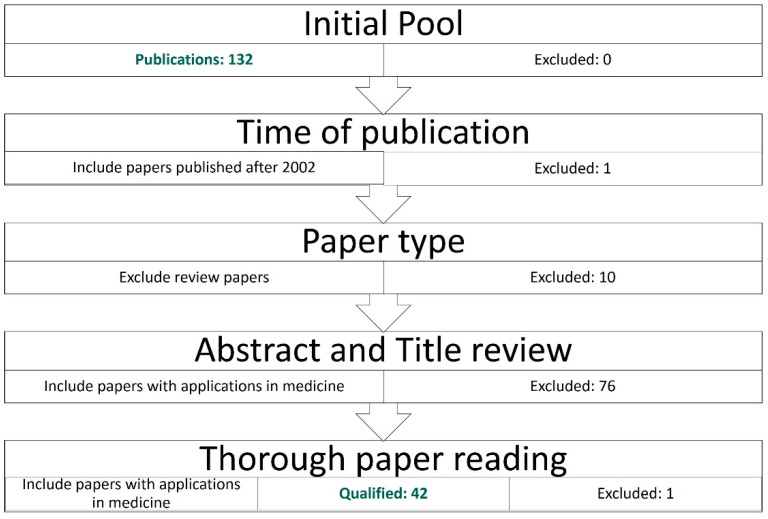
Literature collection process.

**Figure 2 bioengineering-11-00139-f002:**
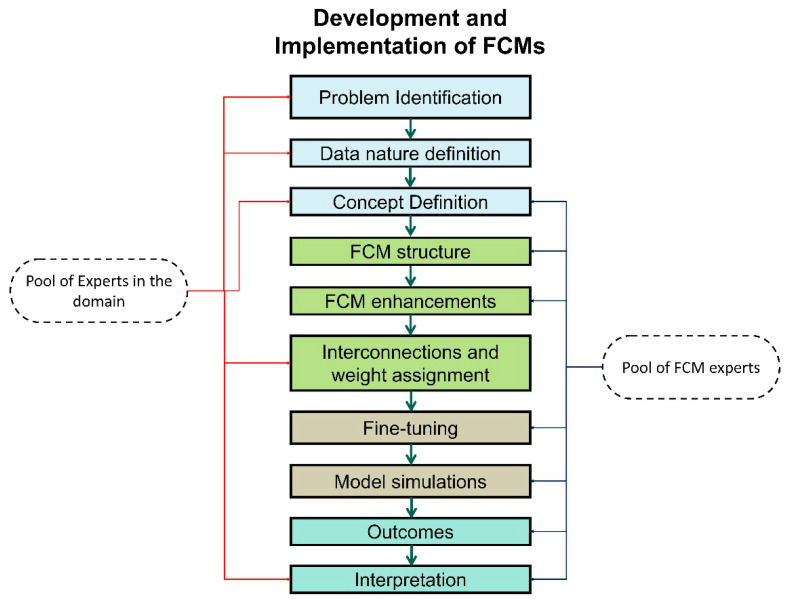
FCM development process.

**Figure 3 bioengineering-11-00139-f003:**
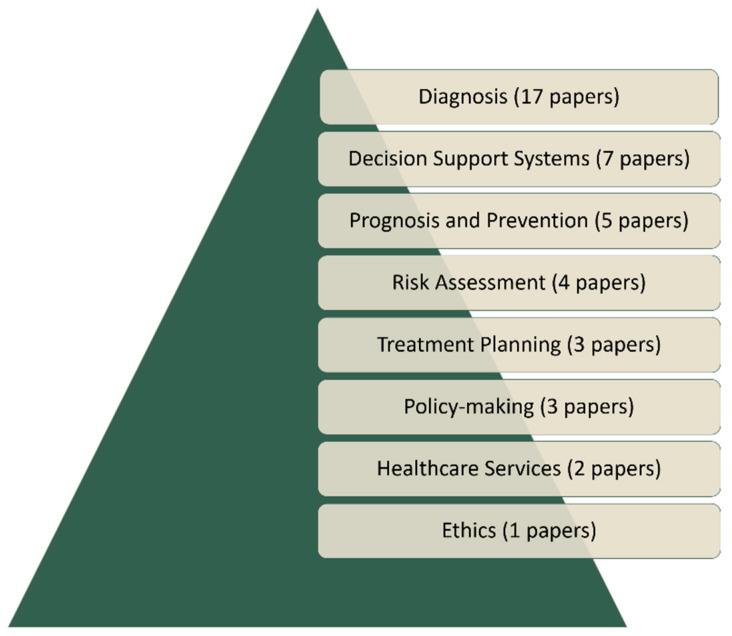
General categories of FCM applications in the medical domain.

**Figure 4 bioengineering-11-00139-f004:**
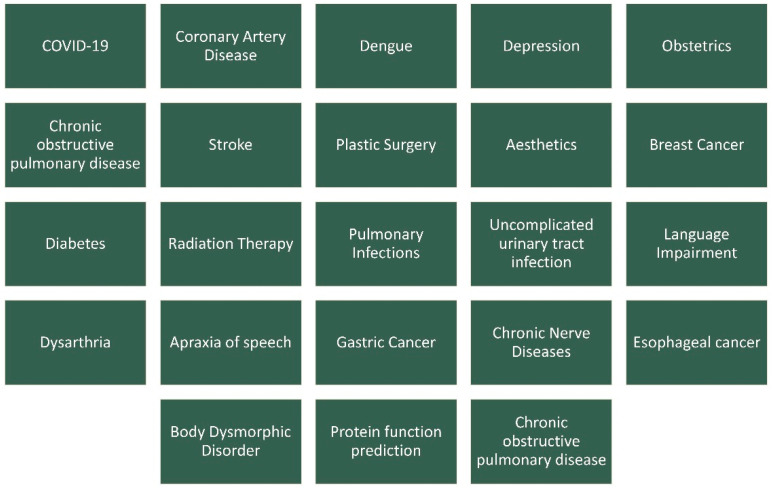
Entities found in the literature concerning the applications of FCMs in medicine.

**Figure 5 bioengineering-11-00139-f005:**
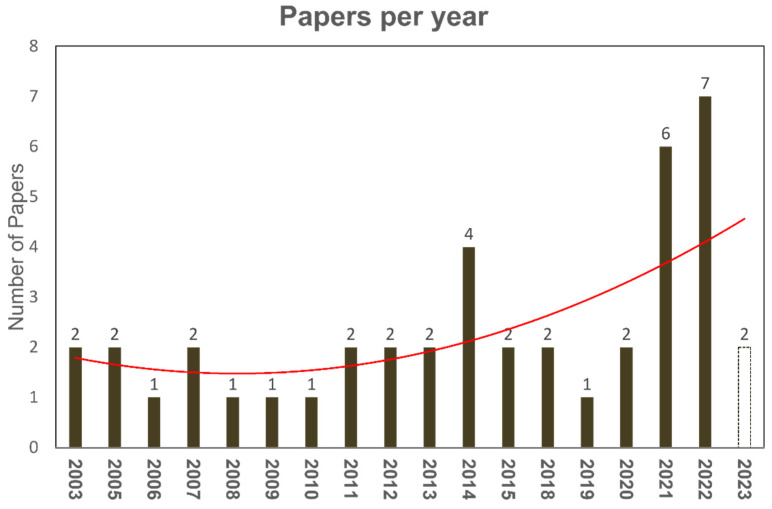
FCM publications per year. The red line capsules the growing trend of publications over the last years.

**Table 2 bioengineering-11-00139-t002:** Summary of findings.

Publication	Key Contribution
Feleki et al. [24]	Proposed DeepFCM for CAD diagnosis, integrating MPI, clinical data, and natural language insights. Achieved 83.07% accuracy, 86.21% sensitivity, and 79.99% specificity. Enhanced explainability with Grad-CAM, weight disclosure, and GPT 3.5.
Feleki et al. [29]	Introduced FCM-PSO and DeepFCM for CAD classification, achieving 77.95% accuracy. The DeepFCM framework combines image and clinical data, providing enhanced explainability for nuclear physicians by revealing meaningful causal relationships between clinical factors in diagnosis. The method outperforms traditional machine learning algorithms, demonstrating efficiency in CAD diagnosis.
Al-Halabi et al. [23]	Analyzed mental models in breast augmentation and flexor tendon repair in plastic surgery. Identified five cognitive competency domains: situation awareness, decision-making, task management, leadership, and communication/teamwork. Framework aids teaching and assessment of competencies.
Hoyos et al. [25]	Clinical decision-support system for dengue diagnosis utilises a fuzzy cognitive map, achieving 89.4% accuracy. The model, based on signs, symptoms, and laboratory tests, provides an explainable method for evaluating dengue severity.
Sovatzidi et al. [26]	Proposed Constructive FCM framework utilises EEG data for adolescent depression severity estimation. Simplifies FCM creation, reduces expert involvement, ensures interpretability, and demonstrates effectiveness on a recent dataset.
Sarmiento et al. [45]	Fuzzy cognitive mapping in northern Nigeria identified frequent sex, lack of contraception use, family dynamics, and lack of male involvement as major causes of short birth intervals (kunika). Cultural dynamics emphasized the need for comprehensive strategies beyond contraception promotion.
Apostolopoulos et al. [10]	FCM for CAD detection uses State Space Advanced FCMs with rule-based mechanism. Achieves 85.47% accuracy, an improvement of more than 7% over traditional FCM.
Apostolopoulos et al. [46]	MDSS for CAD diagnosis using FCMs achieves 78.2% accuracy, surpassing state-of-the-art classification algorithms.
Amirkhani et al. [22]	A novel Fuzzy Cognitive Map (FCM) model predicts DNA-binding residues in protein sequences, achieving AUC = 0.72. Outperforms hybridNAP and various machine learning methods. Enhanced performance attributed to FCM’s intrinsic feature incorporating input feature relations.
Lucchiari et al. [47]	Illustrates a differential diagnosis between psychogenic non-epileptic seizures (PNES) and epileptic seizures (ES). In the model, decision-concepts (PNES, ES) and factor-concepts (important distinguishing factors) are represented, demonstrating the FCM’s applicability in ambiguous contexts with incomplete or unreliable information.
Douali et al. [48]	Case-based fuzzy cognitive maps proposed for medical diagnosis, evaluated against Bayesian belief networks. Utilised a semantic web framework with a database of 174 patients, demonstrating the approach’s effectiveness through statistical comparison.
Lee et al. [49]	Proposed method designs sinusoidal-type and linear activation functions for FCMs in clinical decision-making. Focuses on improving visibility and stability in inference processes, addressing the limitations of sigmoid functions. Applied to pulmonary infections, results indicate appropriateness for clinical decision-making.
Georgopoulos et al. [27]	The GAFI-CFCM hybrid methodology, combining genetic algorithms and FCM, successfully applied to a diagnostic support model. Effectively processes patient information, reaching the most probable disorder diagnosis out of three options.
Nguyen et al. [50]	Utilises protein homologies and interaction network topology to enhance recall in predictions. Successfully annotates protein functions in *Saccharomyces cerevisiae*, *Caenorhabditis elegans*, and *Drosophila melanogaster*. Outperforms four state-of-the-art methods in terms of recall, precision, Matthews correlation coefficient, harmonic mean, and area under the ROC curves.
Papageorgiou et al. [28]	Developed an advanced diagnostic method for urinary bladder tumor grading using FCMs augmented with unsupervised active Hebbian learning (AHL) algorithm. Achieved a classification accuracy of 72.5%, 74.42%, and 95.55% for tumor grades I, II, and III, respectively. The method combines soft computing FCMs with specialised histopathological knowledge and AHL, providing a transparent and explainable solution for physicians.
Georgopoulos et al. [51]	Proposes a soft computing system for the differential diagnosis of dysarthria and apraxia of speech. Utilises a hierarchical FCM approach based on an established dysarthria classification system. The system aims to provide a “second opinion” or training support for clinicians facing the challenging task of accurate diagnosis of speech disorders. Tested successfully using case studies and real patients.
Georgopoulos et al. [52]	FCM-based approach for differentiating Specific Language Impairment (SLI) from dyslexia and autism. Initial phase uses literature-based “experts” for model development, with plans to involve expert specialists for enrichment.
Meier et al. [44]	METHAD employs machine intelligence for ethical advisory in medical dilemmas, using FCMs. The framework systematically breaks down medical ethics cases, employing fuzzy cognitive maps for ethical advisory. While the technology exists, ethical concerns arise, emphasizing the importance of human contact and empathy in medicine.
Poleto et al. [43]	FCMs, relying on expert knowledge, create intelligible cognitive maps and simulate scenarios for understanding and enhancing cybersecurity strategies in telehealth services.
Papageorgiou et al. [53]	A novel evolutionary-based fuzzy cognitive map (FCM) methodology enhances patient state forecasting in pulmonary infections. The research achieves improved prediction efficiency through innovative data fuzzification for observables and optimization of transformation function gain using evolutionary learning. Validation with real patient data from internal care units demonstrates lower prediction errors compared to conventional genetic-based algorithms.
Georgopoulos et al. [30]	Scenario-based Fuzzy Cognitive Maps (FCM) in Medical Decision Support Systems (MDSS) aid training by simulating decision-making processes. Illustrated in labor decision-making, this approach enhances medical professional education for improved patient care.
Papageorgiou et al. [31]	Bayesian belief networks (BBNs) and fuzzy cognitive maps (FCMs) were employed for medical knowledge formalization in decision support within a semantic web framework. A general-purpose reasoning engine, EYE, facilitated reasoning on these models. Validation using the UTI therapy problem demonstrated the reliability and efficiency of these approaches in semantic web decision support tasks.
Lucchiari et al. [32]	Underscores the importance of diagnostic reasoning in clinical practice and critiques the limitations of traditional and solely objective models. It acknowledges the impact of cognitive biases on diagnoses and challenges in implementing debiasing techniques. The proposed solution integrates cognitive understanding with technology, suggesting a conceptual scheme using fuzzy cognitive maps for safer medical practices.
Stylios et al. [15]	A hierarchical Fuzzy Cognitive Map (FCM) in Medical Decision Support Systems integrates factors for dynamic decision-making, enhancing maternal safety. Enables strategic planning and timely decisions, minimizing fetal distress and maternal complications.
Papageorgiou et al. [54]	The proposed FCM-based architecture integrates various data types, enhancing the model through unsupervised learning. FCMs provide robust reasoning by capturing complex relationships among concepts. The approach synergises fuzzy and neural techniques, incorporating rules from knowledge processing and data mining. It ensures transparency and interpretability in medical decision-making.
Papageorgiou et al. [55]	Decision Tree-Fuzzy Cognitive Map hybrid model that enhances decision-making tasks, effectively handling different types of input data, demonstrated within a medical context.
John et al. [56]	Matlab prototype tested for influenza diagnosis using fuzzy cognitive maps. Symptom observations considered duration certainty and intensity variations. Results showed support index variation for differential diagnosis, emphasizing parameter tuning and uncertainty consideration for practical applicability.
Babroudi et al. [41]	Findings highlight hospital reliability, hygiene, and completeness as top influential factors in improving health service quality during infectious disease circumstances.
Groumpos et al. [57]	Advanced Fuzzy Cognitive Maps (AFCM) model to predict the COVID-19 pandemic’s spread. Unlike statistical models, AFCM captures dynamic cause-and-effect relationships among predefined factors. The model, evaluated using data from Greece, South Korea, and Germany, achieved high accuracy in predicting confirmed cases, with coefficients of determination and Pearson’s correlation coefficients demonstrating its effectiveness.
Dogu et al. [42]	The study combines statistical-based fuzzy cognitive maps (SBFCM) and artificial neural networks (ANN) for predicting hospital stay length in COPD patients. SBFCM provides statistical analysis and gathers expert opinions to define input variables for the ANN model. The integrated approach outperforms other methods with 79.95% accuracy, emphasizing the value of expert opinions in medical decision support for enhanced hospital management and better predictions in COPD care.
Saul et al. [58]	Presents FCMs to explore an individual’s personal construction system, identifying barriers to behavior change. Illustrated using a simulated case on healthy habit adoption, it suggests the potential for targeted psychological interventions.
Wu et al. [33]	Extended probabilistic linguistic fuzzy cognitive map model addresses the complexities of rural elderly health in China. Education is identified as the most critical factor influencing rural elderly health, followed by occupational history, psychology, and physical exercise. Intergenerational relationships gain prominence.
Khodadadi et al. [34]	An FCM approach is proposed for ischemic stroke risk diagnosis, employing non-linear Hebbian learning. Neurologists’ opinions determine individual risk rates, achieving a high accuracy of 93.6 ± 4.5% in testing using 110 real cases, outperforming support vector machine and K-nearest neighbors models.
Najafi et al. [35]	A cross-sectional study in Zanjan Province, Iran, explores the hypothesis that food insecurity contributes to esophageal cancer among women. Utilizing fuzzy cognitive maps (FCMs) for analysis, the research involves 580 women, revealing a 23% and 38% prevalence of hunger and hidden hunger, respectively. Only 39% have secure access to key nutrients. The study suggests an association between food insecurity, body mass index (BMI), and esophageal cancer. Results highlight the impact of food insecurity on nutritional status and its potential role in cancer development.
Billis et al. [59]	Modular decision support framework for aging well, including trend analysis, decision support core, and risk prediction. The trend analysis uses personalised sleep models, while the decision support core accurately identifies health states and monitors disease progression. Risk prediction, employing FCM-based approaches, shows promising results, especially in detecting depression. Real-life clinical case testing is crucial for validation.
Mahmoodi et al. [37]	Employed FCMs based on the Nonlinear Hebbian Learning (NHL) algorithm for complex system modeling. Utilised data from the medical records of 560 patients and selected 27 effective features with expert opinions. Achieved a prediction accuracy of 95.83%, surpassing other decision-making algorithms like decision trees, Naïve Bayes, and ANN. The proposed system is deemed simple, comprehensive, and effective for healthcare professionals in predicting gastric cancer risk factors in clinical settings.
Subramanian et al. [36]	Developed an integrated breast cancer risk prediction model using a two-level fuzzy cognitive map (FCM). Combined demographic factors and screening mammogram findings for enhanced risk assessment. Employed Hebbian-based learning to improve the model’s performance and aid in tumor grading and risk prediction. Demonstrated superior accuracy compared to benchmark machine learning methods.
Papageorgiou et al. [38]	Created a familial breast cancer risk assessment model using FCMs, focused on personalised decision-making and incorporating family history and demographic risk factors to identify hidden risks of breast cancer. Utilised Hebbian-based learning capabilities of FCM to enhance modeling and contribute to risk prediction.
de Brito et al. [60]	Developed a fuzzy model to quantify body image dissatisfaction. Successfully measured distress levels in cosmetic surgery patients. The model serves as a screening tool for Body Dysmorphic Disorder BDD in cosmetic surgery. Applicable in psychiatric practice for treating BDD patients.
Papageorgiou et al. [40]	Proposed fuzzy cognitive maps (FCMs) for modeling uUTI treatment decision-making. Developed a software tool, FCM-uUTI DSS, to assist in uUTI treatment management. Evaluated the tool in 38 patient cases, demonstrating reliability and functionality. Results showed the FCM-uUTI tool provides antibiotic suggestions for uUTI treatment. Highlighted the tool’s potential to make medical knowledge widely available through computer consultation systems.
Giles et al. [39]	Used FCMs to compare Aboriginal and conventional science perspectives on diabetes determinants. FCM detailed the complex system of culture, spirituality, and balance in the Aboriginal view. Highlighted tangible stressors and outcomes amenable to management and monitoring. Demonstrated FCM’s potential to integrate diverse perspectives into health management and policy.
Papageorgiou et al. [61]	Introduced FCMs for decision-making in radiation therapy. Used FCMs to estimate the final dose delivered to the target volume. Proposed a two-level integrated hierarchical structure for supervision and evaluation. Applied the methodology to two clinical case studies for testing and evaluation. Discussed the usefulness of the hierarchical structure and suggested future research directions.

## Data Availability

Not applicable.

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
