# Peer review of "Fuzzy Cognitive Map Applications in Medicine over the Last Two Decades: A Review Study"

_bioengineering, 2024, doi:10.3390/bioengineering11020139_

Round 1
Reviewer 1 Report
Comments and Suggestions for Authors
My comments are in the uploaded file.

My comments are in the uploaded file.
Reviewer 2 Report
Comments and Suggestions for Authors
The review article titled "Fuzzy Cognitive Map Applications in Medicine over the Last Two Decades" offers a comprehensive exploration of the applications of Fuzzy Cognitive Maps (FCMs) in the field of medicine. However, there are several points that need to be addressed for further enhancement:
Definition of Review Gap:
The review article fails to clearly define the review gap, leaving the reader without a distinct understanding of why this review contributes significantly to the existing body of literature. To strengthen the manuscript, the authors should explicitly highlight how their work advances beyond previous review studies in the field and articulate the unique contributions this review brings to the domain of FCM applications in medicine.
Methodology for Literature Selection:
The paper does not sufficiently detail the methodology employed for selecting the reviewed works. Considering the nature of a review study, the authors should consider adopting the Preferred Reporting Items for Systematic Reviews and Meta-Analyses (PRISMA) statement as a framework. This would enhance transparency in the literature selection process and provide a standardized approach for future researchers to follow.
Insufficient Result Comparison:
The review lacks a detailed and comprehensive comparison of results presented in the various reviewed studies. A more thorough analysis and synthesis of findings across different works are needed to provide a nuanced understanding of the trends, patterns, and inconsistencies in the applications of FCMs in the medical domain.
Graphical Representation of Results:
The paper should incorporate graphical representations to facilitate a clearer understanding of the comparative results. Visual aids such as charts or graphs could enhance the presentation of findings and contribute to the overall clarity of the review.
Increase Citations for a Comprehensive Review:
The number of cited works in the review appears to be limited. To strengthen the academic rigor and comprehensiveness of the study, the authors should consider including a broader range of relevant literature. This will ensure a more robust foundation for the review and provide readers with a broader perspective on the subject.
Clarification and Unification of Metrics:
The article should offer a more detailed explanation of the metrics used in the reviewed studies. Additionally, if possible, the authors should consider unifying the metrics to facilitate a more meaningful and standardized comparison across different applications of FCMs in medicine.
Addressing these points will significantly enhance the overall quality and impact of the review article, providing a more valuable resource for researchers and practitioners in the field.
Reviewer 3 Report
Comments and Suggestions for Authors
The article entiled Fuzzy Cognitive Map applications in medicine over the last two decades: A review study is well-written and, from my point of view, would be of interest for the readers of Bioengineering. In spite of this and before its publication I consider that authors should take into account the following issues:
In the introductory section, authors should included a description of the layout of the whole manuscript indicating its sections and their content.
Lines 50-51: about the systematic review, any well-known methodoloy has been employed? If so, please cite it in the manuscript. For example, a well-know methodology and largely employed is PRISMA (Preferred Reporting Items for Systematic Reviews and Meta-Analyses).
Figure 1: a more in-depth explanation of the exclusion criteria is required.
Line 134: check the font size of w(i,j)
Figure 3 and figure 4: from my point of view, both are really important for a good understanding of the problem under study but in order to link them with the literature review I consider it would be of interest to indicate how many articles lay in any of the categories it would complement the information of Table 1. About Figure 5, i am not sure if including year 2023 is covenient or not as the research was closed before the year finished. If included please indicate clearly in the text and draw the bar in a different way (how about dot lines?)
Round 2
Reviewer 1 Report
Comments and Suggestions for Authors
Looks good.
Reviewer 3 Report
Comments and Suggestions for Authors
After the changes performed by the authors, the article is ready for its publication. Congratulations.